# On the CCN [de]activation nonlinearities

**Sylwester Arabas**[1,2] **and Shin-ichiro Shima**[3]

[1]Institute of Geophysics, Faculty of Physics, University of Warsaw, Warsaw, Poland
[2]Chatham Financial Corporation Europe, Cracow, Poland
[3]Graduate School of Simulation Studies, University of Hyogo, Kōbe, Japan

*Correspondence to:* Sylwester Arabas (sarabas@chathamfinancial.eu) and Shin-ichiro Shima (s_shima@sim.u-hyogo.ac.jp)

**Abstract.** We take into consideration the evolution of particle size in a monodisperse aerosol population during activation and deactivation of cloud condensation nuclei (CCN). Our analysis reveals that the system undergoes a saddle-node bifurcation and a cusp catastrophe. The control parameters chosen for the analysis are the relative humidity and the particle concentration. An analytical estimate of the activation timescale is derived through estimation of the time spent in the saddle-node bifurcation bottleneck. Numerical integration of the system coupled with a simple air-parcel cloud model portrays two types of activation/deactivation hystereses: one associated with the kinetic limitations on droplet growth when the system is far from equilibrium, and one occurring close to equilibrium and associated with the cusp catastrophe. We discuss the presented analyses in context of the development of particle-based models of aerosol-cloud interactions in which activation and deactivation impose stringent time-resolution constraints on numerical integration.

## 1 Background

Atmospheric clouds are visible to human eye for they are composed of water and ice particles that effectively scatter solar radiation. The multi-micrometre light-scattering cloud droplets form on sub-micrometre aerosol particles (cloud condensation nuclei, CCN) in a process referred to as CCN activation or (heterogeneous) nucleation. The concentration (from tens to thousands per cm$^{-3}$) and size (from fractions of to multiple micrometres) of activated particles can both vary by over an order of magnitude depending on the size spectrum and composition of CCN. On one hand, CCN physicochemical properties are influenced by anthropogenic emissions of particles into the atmosphere. On the other hand, the resultant size spectrum of cloud droplets determines how effectively the clouds interact with solar radiation and how effectively they produce precipitation (see e.g. a recent NPG paper by Feingold and Koren, 2013, for a discussion of aerosol-cloud-precipitation interaction chain, unconventionally modelled as a predator-prey problem). CCN activation is thus the linking process between the microscopic human-alterable atmospheric composition and the macroscopic climate-relevant cloud properties. As once aptly stated, "*there is something captivating about the idea that fine particulate matter, suspended almost invisibly in the atmosphere, holds the key to some of the greatest mysteries of climate science*" (Stevens and Boucher, 2012). This has certainly contributed to the wealth of literature on the subject published since the first studies of the 1940-ties (Howell, 1949; Tsuji, 1950), for a thorough list of references see e.g., Khvorostyanov and Curry (2014, chpt. 7).

Deactivation is the reverse process in which cloud droplets evaporate back to aerosol-sized particles. The process is also referred to as aerosol regeneration, aerosol recycling, drop-to-particle conversion or simply droplet evaporation (see section 1 in Lebo and Seinfeld, 2011, for a review of modelling studies). Both activation and deactivation are particular cases of particle condensational growth which, in context of cloud modelling, is generally regarded as reversible to contrast the irreversible collisional growth (see e.g., Grabowski and Wang, 2013). The reversibility of condensational growth is a sound (and often a constituting) assumption for cloud models for which activation and deactivation are subgrid processes, both in terms of time- and length-scales. Yet, when investigated in short-enough timescales, condensation and evaporation exhibit a hysteretic behaviour in an activation-deactivation cycle. The hysteresis can be associated with the kinetic limitations in the vapour and heat transfers to/from the droplets (Chuang et al., 1997) and has been previously depicted in the studies of Korolev and Mazin (2003, discussion of Fig. 1), Pinsky et al. (2013) and Korolev et al. (2013).

As we point out in this note, the system can exhibit a hysteretic behaviour also in a close-to-equilibrium régime where the kinetic limitations do not play a significant role.

It is worth noting that particle nucleation through condensation is relevant in a much wider context than formation of atmospheric clouds. Since late 19th century, the growth of particles through condensation up to optically-detectable sizes has been the principle of operation of so-called condensation particle counters (see McMurry, 2000, for a historical review). Instruments in which single CCN undergo activation in conditions similar to those discussed herein (Roberts and Nenes, 2005) are routinely used in ground-based and airborne research measurements. Interestingly, an analogue of CCN activation theory applies to the formation of nanometre-size aerosol particles via activation of molecular clusters by organic vapours (Kulmala et al., 2004).

The note is structured as follows. Section 2 provides a brief introduction to the constituting elements of the CCN activation theory. In sections 3–5, we detail how the dynamics of cloud droplet growth can be studied employing the techniques of nonlinear dynamics analysis. In these sections we do not refrain from using the peculiar yet pertinent jargon of nonlinear dynamics. For introduction, we refer the reader to the concise and approachable introductory chapters in Strogatz (2014, chapters 2.0–2.2, 2.4, 3.0) as well as to sections on specific topics therein to which references are provided throughout the text. Sections 6–7 deal with the so-called *air parcel* cloud model framework. The framework is used here to corroborate the results from nonlinear dynamics analysis of a simplified CCN activation model against numerical solutions of an equation system providing more comprehensive description of the process. Section 8 provides an additional context for the discussion by pointing out the congruence of the simplifying assumptions embraced in the presented analysis with the recently popularised particle-based techniques for modelling aerosol-cloud interactions. Summary section 9 concludes the note.

## 2 Droplet growth laws in a nutshell

The key element in the mathematical description of CCN activation/deactivation is the equation for the rate of change of particle radius $r_w$ (so-called wet radius) due to water vapour transfer to/away from the particles. It is modelled by a diffusion equation in a spherical geometry,

$$\dot{r}_w = \frac{1}{r_w} \frac{D_{eff}}{\rho_w} (\rho_v - \rho_\circ) \; , \tag{1}$$

where $\rho_w$ is the liquid water density, $\rho_v$ is the ambient vapour density (away from the droplet), $\rho_\circ$ is the equilibrium vapour density at the drop surface and the $D_{eff} = D_{eff}(T, r_w)$ is an effective diffusion coefficient in which the temperature dependence stems from an approximate combination of the Fick's first law and Fourier's law (latent heat release) into a single particle-growth equation (so-called Maxwell-Mason formula), while the radius dependence stems from corrections limiting the diffusion efficiency for smallest particles. For derivation and discussion see Khvorostyanov and Curry (2014, section 5.1.4). Introducing two non-dimensional numbers: the relative humidity $RH = \rho_v/\rho_{vs}$ (the ratio of the ambient vapour density to the vapour density at saturation with respect to plane surface of pure water) and the equilibrium relative humidity $RH_{eq} = \rho_\circ/\rho_{vs}$, the drop growth equation is given by

$$\dot{r}_w = \frac{1}{r_w} D_{eff} \frac{\rho_{vs}}{\rho_w} (RH - RH_{eq}) \; . \tag{2}$$

The crux of the matter is the dependence of $RH_{eq}$ on $r_w$. In the context of atmospheric clouds, it is determined primarily by the droplet curvature and by the presence of dissolved substances. The theory capturing the interplay between these two effects was formulated by Köhler in 1936. To note, the qualitatively similar interplay between the surface tension and electric charge (as opposed to chemical composition) results in an analogous particle activation phenomenon which served as the principle of operation of the Wilson cloud chamber – a key instrument in the early days of elementary particle physics (for references, see McMurry, 2000).

The Köhler theory provides us with the so-called Köhler curve, the leading terms of its common $\kappa$-Köhler formulation can be approximated with (for $r_d \ll r_w$ which is a reasonable assumption in context of activation/deactivation)

$$RH_{eq} = \frac{r_w^3 - r_d^3}{r_w^3 - r_d^3(1-\kappa)} \exp\left(\frac{A}{r_w}\right) \tag{3}$$

$$\approx 1 + \frac{A}{r_w} - \frac{\kappa r_d^3}{r_w^3} \; , \tag{4}$$

where $A \sim 10^{-3} \mu m$ is a temperature-dependant coefficient related with surface tension of water, while the dry radius $r_d$ and the solubility parameter $\kappa$ (in general, $0 < \kappa < 1.4$, see Petters and Kreidenweis, 2007) are proxy variables depicting the mass and chemical composition of the substance the CCN are composed of. The $\partial_{r_w} RH_{eq}$ derivative has an analytically-derivable root corresponding to the maximum of the Köhler curve at $(r_c, RH_c)$ where $r_c = \sqrt{3\kappa r_d^3/A}$ is the so-called critical radius and $RH_c = 1 + \frac{2A}{3r_c}$ is the critical relative humidity.

## 3 Saddle-node bifurcation at Köhler curve maximum

Rewriting equation 2 in terms of $\xi = r_w^2 + C$ (where $C$ is an arbitrary constant) gives

$$\dot{\xi} = 2 D_{eff} \frac{\rho_{vs}}{\rho_w} (RH - RH_{eq}(\xi)) \; . \tag{5}$$

Figure 1 depicts the phase portrait of the dynamical system defined by (5), for different values of relative humidity

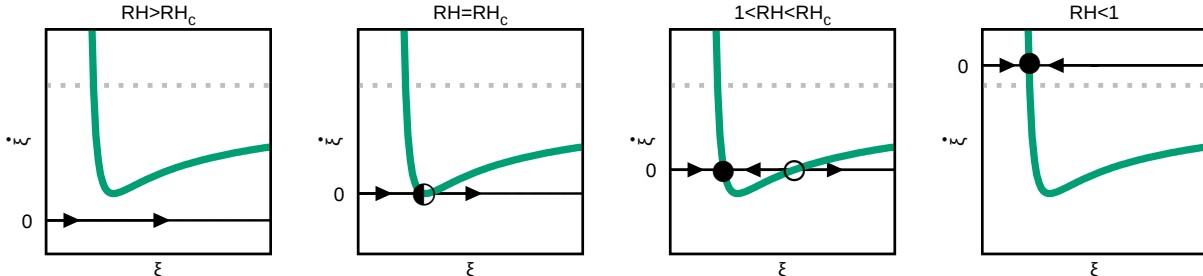

**Figure 1.** Phase portraits of the system discussed in Section 3 for different values of the control parameter RH. Arrows have their heads pointing right (left) if the sign of $\dot{\xi}$ is positive (negative). Half-filled circle denotes a half-stable fixed point. Filled and open circles denote stable and unstable fixed points, respectively. Dashed line corresponds to RH = 1.

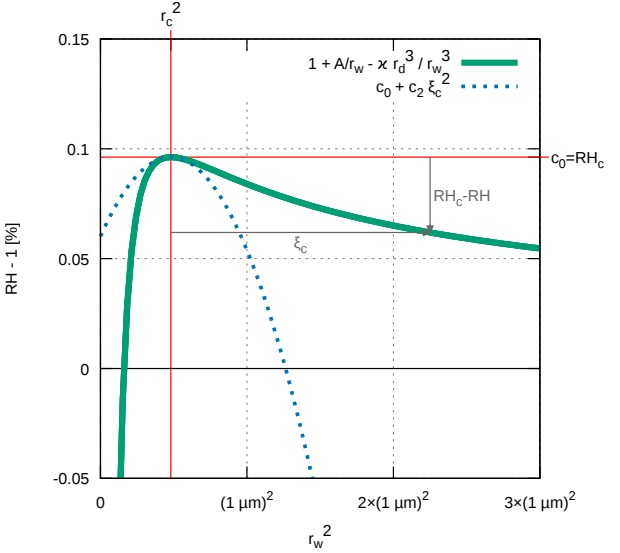

**Figure 2.** Köhler curve for CCN with $r_d = 0.05 \mu m$, $\kappa = 1.28$ (NaCl) and its Taylor expansions at $r_c$ and at infinity.

RH which is chosen as the control parameter in the following fixed point analysis. Fixed points correspond to equilibrium conditions defined by $\dot{\xi} = 0$ which can be geometrically identified as crossings of the $-\mathrm{RH_{eq}}$ curve (a flipped Köhler curve) and the constant function RH.

For $\mathrm{RH} > \mathrm{RH_c}$, there are no intersections – there are no fixed points, the time derivative $\dot{\xi_c}$ is always positive: regardless of their size, the CCN grow. For $\mathrm{RH} = \mathrm{RH_c}$, there is just one fixed point, it is half-stable (small variation in $\xi$ can be either damped or amplified depending on the direction). For $1 < \mathrm{RH} < \mathrm{RH_c}$, there are two fixed points in the system: one stable and one unstable. The stability depends on how the sign of $\dot{\xi}$ changes around a fixed point (note that the arrows on the plot correspond to the sign of $\dot{\xi}$). Around a stable fixed point (also called attractor, sink), small variations in $\xi$ are damped, while in the case of unstable fixed point (also called repeller, source), small variations in $\xi$ are amplified. Particles smaller in radius than the radius corresponding to the

unstable fixed point will shrink or grow in the direction of the equilibrium state corresponding to the stable fixed point (as depicted by the directions of the arrows). Particles larger in radius than the radius corresponding to the unstable fixed point will grow provided $\mathrm{RH} > 1$ – these are the activated CCN. In the limit of $\xi \to \infty$, the Köhler curve approaches $\mathrm{RH} = 1$, hence the unstable fixed point goes to infinity. For $\mathrm{RH} < 1$, there is just one stable fixed point corresponding to the unactivated CCN equilibrium.

The above analysis portrays a bifurcation in the behaviour of the system at $\mathrm{RH} = \mathrm{RH_c}$. Rewriting $\mathrm{RH_{eq}}$ in terms of $\xi_c = r_w^2 - r_c^2$ and Taylor-expanding it around $\xi_c = 0$ gives:

$$\mathrm{RH_{eq}}(\xi_c) = c_0 + \cancel{c_1 \xi_c} + c_2 \xi_c^2 + \dots \qquad (6)$$

where $c_0 = \mathrm{RH_c}$, $c_1$ is zero as we are expanding around the root of $\partial_{\xi_c} \mathrm{RH_{eq}}$ and $c_2 = -\frac{A}{4} r_c^{-5}$ is negative.

Combining equations 5 and 6 gives

$$\dot{\xi_c}\Big|_{\xi_c \to 0} \sim \frac{\mathrm{RH} - \mathrm{RH_c}}{A/(4r_c^5)} + \xi_c^2 \ , \qquad (7)$$

which is the normal form of the saddle-node bifurcation (Strogatz, 2014, section 3.1).

Noteworthy, the standard cloud-physics Köhler curve plot given in Figure 2 can well serve as a (flipped) phase portrait of the system facilitating identification of the fixed points by considering intersections of the Köhler curve with lines of constant RH. Figure 2 depicts the approximation (7) alongside the kappa-Köhler curve, confirming that the parabolic approximation is valid only in the nearest vicinity of $(r_c, \mathrm{RH_c})$.

## 4 Activation timescale estimation

Interestingly, the analysis of the CCN activation/deactivation in terms of saddle-node bifurcation provides a way to estimate the timescale of activation. Following Strogatz (2014, section 4.3), the *coalescence* of the fixed points is associated with a passage through a *bottleneck*. The key observation is

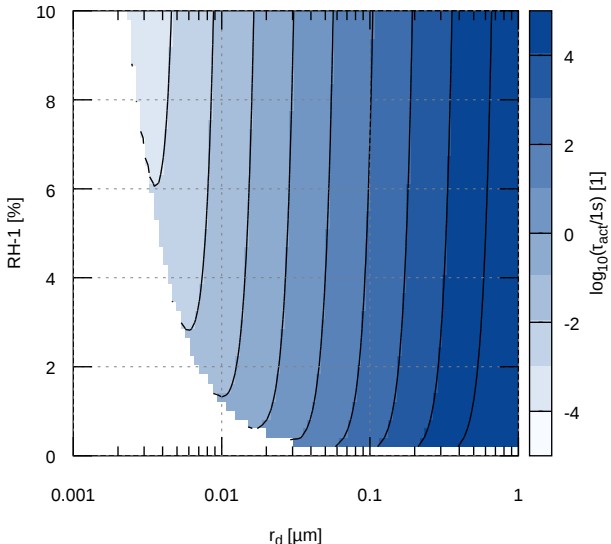

**Figure 3.** Activation timescale as a function of dry radius and relative humidity estimated with equation 8 with $A \sim 10^{-3}\mu m$, $\kappa = 1.28$, $D \sim 2 \times 10^{-5}\frac{m^2}{s}$, $\rho_w \sim 10^3 \frac{kg}{m^3}$ and $\rho_{vs} = 10^{-3}\frac{kg}{m^3}$.

that for the parabolic normal form of the saddle-node bifurcation, the time of the passage through the *bottleneck* dominates all other timescales. Thus, the timescale of the process can be estimated by integrating $\xi_c$ from $-\infty$ to $\infty$:

$$\tau_{act} \approx \int\limits_{-\infty}^{+\infty} \frac{d\xi_c}{\dot{\xi}_c} = \frac{r_c^{5/2}}{\sqrt{A}} \frac{\rho_w/\rho_{vs}}{D_{eff}} \frac{\pi}{\sqrt{RH-RH_c}} \ . \qquad (8)$$

The activation timescale $\tau_{act}$ given by eq. 8, plotted as a function of RH and $r_d$ (and substituting $r_c$ and $RH_c$ by their analytic formulae given in the preceding section) is presented in Figure 3. It matches remarkably the data obtained through numerical calculations presented in Hoffmann (2016). The white region in the plot corresponds to situation where activation does not happen. The range of RH depicted in the plot is chosen to match the one of Figure 2 in Hoffmann (2016), while in principle the presented weakly non-linear analysis of the system is applicable only close to the equilibrium (i.e., close to the edge of the white region in the plot).

## 5 Cusp catastrophe of the RH-coupled system

The key limitation of the preceding analysis is that the evolution of particle size is not coupled with the evolution of ambient heat and moisture content, and hence the relative humidity. Limiting the analysis to a monodisperse population, the coupling efficiency is determined by the total number of particles in the system. The so-far assumed constant RH approximates thus the case of small number of droplets.

To at least partially lift the constant-RH assumption, while still allowing for a concise analytic description of the sys-

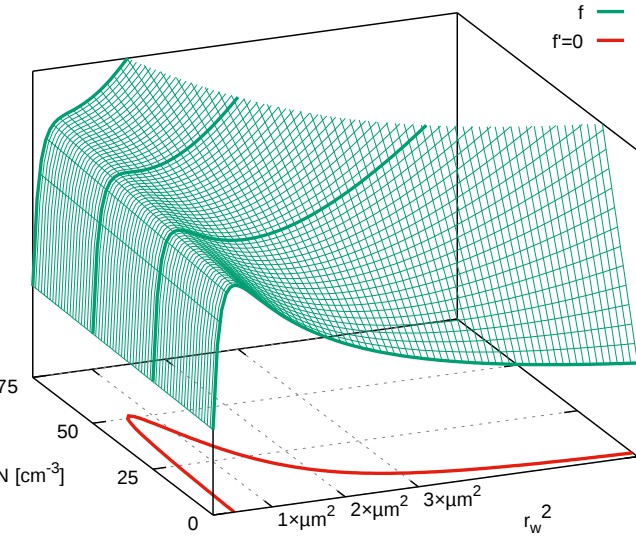

**Figure 4.** Dependence of $f$ defined in eq. 11 on the wet radius and particle concentration (green wireframe surface). Red line below depicts the zero-crossings of the first derivative of $f$ with respect to $r_w$. Values of all constants as in Fig. 3. Discussion in section 5.

tem dynamics, let us consider a simple representation of the moisture budget in the system under a temporary assumption of constant temperature and pressure (and hence constant volume, constant $\rho_{vs}$, $A$ and $D_{eff}$). The rate of change of the ambient relative humidity $\dot{RH}$ can be expressed then as a function of the droplet volume concentration $N$,

$$\dot{RH} \approx \frac{\dot{\rho_v}}{\rho_{vs}} = -N \underbrace{\frac{4\pi \rho_w}{3\rho_{vs}}}_{\alpha} 3r_w^2 \dot{r}_w \ , \qquad (9)$$

where the form of $\alpha$ stems from defining the density of liquid water in the system as $N\rho_w \frac{4}{3}\pi r_w^3$. Integrating in time gives

$$RH = RH_0 - \alpha N r_w^3 \ , \qquad (10)$$

which combined with eq. 2 and expressed in terms of $\xi$ with $C = 0$ leads to the following phase portrait of the RH-coupled system (assuming $r_w \gg r_d$):

$$\dot{\xi} \sim (RH_0 - 1) - \underbrace{\left( \frac{A}{\xi^{\frac{1}{2}}} - \frac{\kappa r_d^3}{\xi^{\frac{3}{2}}} + \alpha N \xi^{\frac{3}{2}} \right)}_{f} \ , \qquad (11)$$

where the group of terms labelled as $f$ can be intuitively thought of as corresponding to the Köhler curve with an additional term representing the simplified RH dynamics.

Figure 4 depicts the dependence of $f$ on the droplet radius $r_w = \sqrt{\xi}$ and the droplet concentration $N$. To facilitate analysis, the zero-crossings of the first derivative of $f$ with respect to $r_w$ are plotted as well using the analytically derived formula

$$\text{sgn}(f') = \text{sgn}\left( \kappa r_d^3 - \frac{A}{3}r_w + \alpha N r_w^3 \right) \ . \qquad (12)$$

For $N = 0$, $f$ has the Köhler-curve shape depicted in Fig. 2 which, as discussed in the preceding sections, implies a saddle-node bifurcation. With $N$ greater than zero but less than ca. 50/cm³, a second saddle-node point appears as the $\alpha N \xi^{\frac{3}{2}}$ term causes $f$ to have a local minimum above the critical radius. At ca. N=50/cm³, both the first and second derivatives of $f$ vanish implying a cusp point in the $f$ surface. For larger $N$, $f$ is monotonic, hence both of the saddle-node bifurcations cease to exist. For $N > 0$, this phase portrait reveals a topological equivalence (cf. Meiss, 2017, theorem 4.3) to the normal form of the cusp bifurcation. The cusp bifurcation (Kuznetsov, 2004, chpt. 8.2), an imperfect supercritical pitchfork bifurcation (Strogatz, 2014, chpt. 3.6), features a cusp catastrophe what allows to envision a "catastrophic" jump from one equilibrium to another and a hysteretic behaviour of the system when approaching (in terms of $r_w$) the local minimum of $f$ from below (activation) and from above (deactivation) for small enough $N$.

## 6 Adiabatic vertically-displaced air parcel system

In order to lift the assumptions of constant temperature and pressure, the system evolution can be formulated by supplementing the drop growth equation with two equations representing the hydrostatic balance and the adiabatic heat budget. This leads to a commonly used so-called air-parcel framework depicting behaviour of a vertically displaced adiabatically isolated mass of air:

$$\begin{bmatrix} \dot{p}_d \\ \dot{T} \\ \dot{r}_w \end{bmatrix} = \begin{bmatrix} -\rho_d g w \\ (\dot{p}_d/\rho_d - \dot{q} l_v)/c_{pd} \\ (\text{Eq. 1}) \end{bmatrix} \quad (13)$$

where $\rho_d$ and $p_d$ are the dry-air (background state) density and pressure, $w$ is the vertical velocity of the parcel, $q = \rho_v/\rho_d$ is the water vapour mixing ratio, $c_{pd}$ is the specific heat of dry air, $l_v$ is the latent heat of vapourisation and $g$ is the acceleration due to gravity. The sum of water vapour and liquid water densities is conserved in the system which allows to diagnose $\rho_v$ and RH from the state variables, similarily as in eqs. (9)-(10) but without the simplifying assumption of constant temperature and pressure.

As discussed in section 5, for a monodisperse population of $N$ particles, the changes in the mass of liquid water in the system are proportional to the particle concentration, hence $\dot{q} \sim N$. Consequently, the analysis of the activation/deactivation dynamics presented in sections 3-4 under the assumption of constant RH corresponds to the behaviour of the air-parcel system defined by eq. 13 in the limit of:

 – $w \to 0$ (and hence $\dot{p}_d \approx 0$) i.e., slow, close-to-equilibrium evolution of the system relevant to fixed-point analysis (by some means pertinent to the formation of non-convective clouds such as fog) and

 – $N \to 0$ (and hence $\dot{r} \approx 0$) i.e., weak coupling between particle size evolution and the ambient thermodynamics (pertinent to the case of low particle concentration).

## 7 Numerical simulations

Since the system defined by (13) is less susceptible to a simple analytic analysis, we proceed with numerical integration. Furthermore, employing numerical integration allows to evaluate the Köhler curve in unapproximated form (4) to corroborate the findings obtained with the assumption of $r_d \ll r_w$. To this end, a numerical solver was implemented using the *libcloudph++* library (Arabas et al., 2015) and the CVODE adaptive-timestep integrator (Hindmarsh et al., 2005). Numerical integration is carried out for a system equivalent to (13) but expressed in terms of the state variables used in *libcloudph++*: water vapour mixing ratio, potential temperature and wet radius (see Appendix A in Arabas et al., 2015); supersaturation $S = RH - 1$ is diagnosed from the three state variables. The solver code is free and open-source and is available as an electronic supplement to this note.

In order to depict an activation-deactivation cycle, the vertical velocity $w$ was set to a sinusoidal function of time $t$ such that the maximal displacement is reached at $t = t_{\text{hlf}}$ and the average velocity is $<w>$:

$$w = <w> \frac{\pi}{2} \sin\left(\pi \frac{t}{t_{\text{hlf}}}\right). \quad (14)$$

Figure 5 summarises results of nine simulations in three types of coordinates: displacement vs. supersaturation (the top row), supersaturation vs. wet radius (the middle row, same coordinates as in Fig. 2) and displacement vs. wet radius (bottom row). The nine model runs correspond to three sets of aerosol parameters (left, middle and right columns) and three values of mean vertical velocity (depicted by line thickness). The varied aerosol input parameters are the concentration ($N_{\text{STP}}$ of 50 and 500 cm⁻³, STP subscript corresponding to the values at standard temperature and pressure) and the dry radius ($r_d$ of 0.1 and 0.05 μm). In all panels, black lines correspond to air-parcel ascent (activation) and orange lines correspond to the descent (deactivation). Besides integration results, the panels in the middle row feature the Köhler curve plotted with thick grey line in the background.

The plots depict that for mean velocities of 100 cm/s and 50 cm/s activation and deactivation are not symmetric and happen far from equilibrium (the Köhler curve). This type of hysteresis corresponds to the kinetic limitations on the transfer of water molecules to/from the droplet surface what prevents the droplets from attaining equilibrium under rapidly changing ambient conditions.

At much lower velocity of 0.2 cm/s, the processes are symmetric and match the equilibrium curve, but only for the $N = 500$ cm⁻³ and $r_d = 0.1$ μm (middle column). A twofold decrease of the dry radius (right column) as well as a tenfold

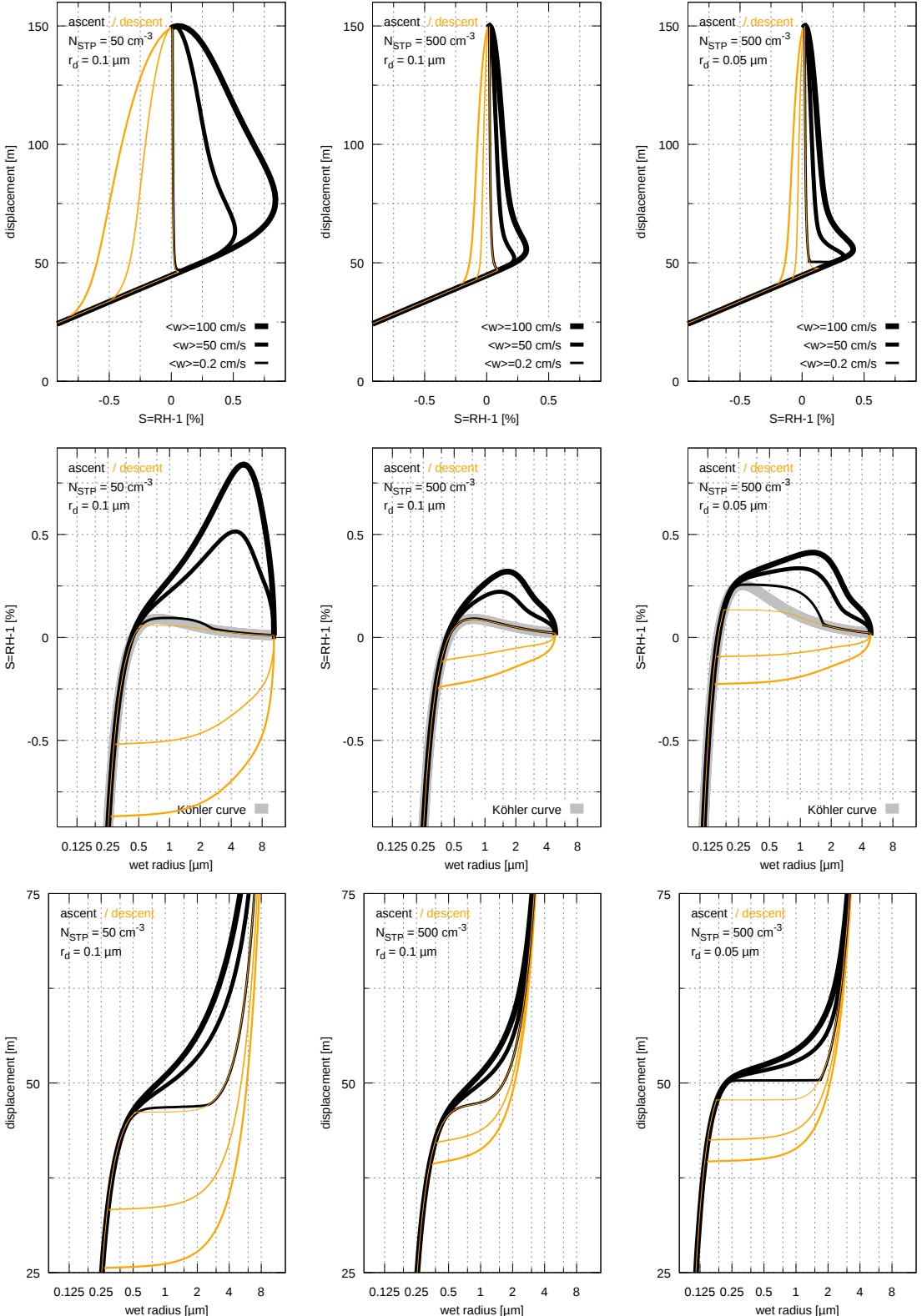

**Figure 5.** Results of numerical simulations discussed in section 7.

decrease of particle concentration (left column) both cause the system to exhibit a hysteretic behaviour also at the lowest considered velocity. This hysteresis is characterised by a "jump" in the wet radius that qualitatively matches the envisioned catastrophic behaviour associated with the cusp bifurcation. This behaviour is robust to further reduction in the vertical velocity (not shown) confirming a close-to-equilibrium régime was attained.

The adaptive-timestep solver statistics (not shown) reveal that regardless of the chosen relative accuracy, for all considered input parameters, there are two instants for which the solver needs to significantly reduce the timestep: when resolving the supersaturation maximum during activation and when resolving the "jump" back to equilibrium during deactivation. It is a robust feature that deactivation requires roughly an order of magnitude shorter timestep as compared to activation (ca. 0.01 s vs. 0.1 s for a relative accuracy of $10^{-6}$). The only exception from this rule is the symmetric case which does not feature the "jump" back onto the equilibrium curve.

## 8   Monodisperse system: limitations and applicability

The key advantage of the embraced monodisperse simulation is simplicity – in terms of model formulation, result analysis but also integration. Due to the wide span of aerosol and droplet size spectrum, simulations of the particle size spectrum evolution during activation are prone to numerical difficulties – both due to the stiffness of the system and due to the sensitivity to the size spectrum discretisation (Arabas and Pawlowska, 2011).

The key inherent limitation for applicability of monodisperse simulations is the lack of description of the cloud droplet size spectrum shape. Consequently, the model lacks representation of the phenomena that depend on simultaneous presence of both activated and unactivated CCN. Such phenomena include the noise-induced excitations to which even a bi-disperse system would be susceptible if subject to fluctuations in the forcing terms (e.g. in the cooling rate $\dot{T}$, see Hammer et al., 2015, discussion of Fig. 10-11 and other studies referenced therein). The excitations influence the partitioning between activated and unactivated CCN, and decay when the characteristic timescale (period) of fluctuations is largely longer or shorter than the activation timescale discussed in section 4.

These limitations certainly restrain the relevance of the presented calculations to real-world problems. Yet, let us underline that both the monodisperse spectrum and even the no-RH-coupling assumption are in fact contemporarily used in atmospheric modelling in the recently popularised particle-based (Lagrangian, super-droplet) techniques for representing aerosol, cloud and precipitation particles in models of atmospheric flows (see e.g., Shima et al., 2009; Arabas and Shima, 2013, as well as works referred therein).

In these models, in the spirit of the particle-in-cell approach, the liquid water is represented with computational particles, each representing a multiplicity of real-world particles with monodisperse size. In such models, the particles can undergo repeated activation-deactivation cycles, potentially also at low vertical velocities. Consequently, the close-to-equilibrium catastrophic hysteresis observed in the presented simulations, even if of no foreseeable relevance to the macroscopic behaviour of the large-scale cloud systems modelled with the particle-based techniques, has to be taken into account when developing numerical integration schemes.

## 9   Concluding remarks

With this note we intend to bring attention to the presence of nonlinear peculiarities in the equations governing CCN activation and deactivation, namely a saddle-node bifurcation and a cusp catastrophe. We have shown that conceptualisation of the process in terms of bifurcation analysis yields a simple yet practically-applicable description of the system allowing analytic estimation of the timescale of activation. Both through weakly-nonlinear analysis and through numerical integration, we have depicted the presence of a cusp catastrophe in the system and the corresponding hysteretic behaviour near equilibrium (i.e., for very small air-parcel velocities).

The deactivation stage was observed to determine the timestepping constraints for numerical integration when simulating an activation-deactivation cycle of a monodisperse droplet population. It is a finding of interest for cloud modelling community since monodisperse activation/deactivation models of the studied type play a constituting role in the more-and-more widespread particle-based models of aerosol-cloud interactions.

*Acknowledgements.* We thank Hanna Pawłowska, Ahmad Farhat as well as three anonymous reviewers for their comments to the initial version of the manuscript. SA acknowledges support of the Poland's National Science Centre (Narodowe Centrum Nauki) [decision no. 2012/06/M/ST10/00434]. This research was supported by JSPS KAKENHI Grant-in-Aid for Scientific Research(B): (Proposal number: 26286089), and by the Center for Cooperative Work on Computational Science, University of Hyogo. This study was carried out during a research visit of SA to Japan supported by the University of Hyogo. SA extends special thanks to Asada and Okamoto families.

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
