# Peer review of "On the CCN [de]activation nonlinearities"

_Nonlinear Processes in Geophysics, 2016_

## Referee Comment (RC1) · Anonymous Referee #1 · 13 Dec 2016

A review of the manuscript npg-2016-50 "On the CCN [de]activation nonlinearities" authored by Sylwester Arabas and Shin-ichiro Shima.

The paper is dedicated to stability and bifurcation analysis of diffusion growth equation in its complete form, which includes curvature term and chemical composition term. Authors show that critical point on the Kohler curve is the point of saddle-node bifurcation. The time scale associated with this point which can be interpreted as time scale of aerosol activation is estimated analytically. Also the analysis of the more complicated case when the aerosol growth is accompanied by decrease of surrounding humidity has been done. The second saddle-node point was found in this analysis. At last the numerical calculation of hysteresis profiles of supersaturation and aerosol wet radius in adiabatic vertically-oscillating air parcel has been done. The article is interesting and I recommend it for publication in NPG after two additions. Firstly I recommend

adding some elementary mathematical explanation what saddle-node bifurcation and cusp catastrophe are. May be authors will write a separate Appendix on this topic. This will help the reader to better understand the article. Secondly I recommend to compare the results obtained in Sections 6, 7 and Fig.4 with results of study by Pinsky et al., (2013: J. Atmos. Sci., 70, 2778-2793). In this study some analytical investigation of monodisperse droplet spectra evolution as well as parcel model investigation are carried out. Possibly the matching of equations from this study and the ones in the reviewed one can bring some new results.

---

## Referee Comment (RC2) · Anonymous Referee #2 · 26 Dec 2016

The authors discuss cloud droplet activation and deactivation in terms of saddle-node bifurcation.

By doing so they can employ a nonlinear dynamic approach to study the properties of such processes. Specifically, they approximate the drop growth by diffusion equation to a normal form of a saddle-node bifurcation ($df/dt = a + f^2$) and therefore they could use the properties of such form to study hysteresis and catastrophe behavior.

The approach and the mathematical insights are very interesting but in order to make this paper accessible to readers from cloud physics, the authors should be much more generous in the details they provide in the mathematical derivations, and invest efforts in translating the mathematical insights to physical ones. A reader that is not fully updated with the jargon of nonlinear dynamics will find this paper hard to follow. The

explanations in some sections are fully based on such jargon (see part 4 - "... the co-alescence of the fixed points is associated with a passage through a bottleneck ...) On the same note, throughout the paper the explanations are very slim. It starts from the overview in the introduction in which the Kohler theory is hardly mentioned (although it is central in the paper). Moreover, many of the numbers provided there are not so accurate. One can have coarse mode aerosols larger than a micron. Concentration can vary between 10's to 10000's etc.

In the next chapters where they develop the mathematical framework, they should add guidelines and physical insights in each of the main steps. What does it mean Saddle-node bifurcation at Köhler curve maximum? They should explain in not from a mathematical point of view (" . . . when the fixed points coalesce into a half-stable fixed point . . .") but from a physical point of view. What is the meaning of this point. What can we learn about it from the Köhler theory?

This is true to all mathematical steps in the paper. While the mathematical derivations look right (as much as I checked) the math derivations details are slim and there is hardly no physical interpretation to the shown insights (which could make this paper much more relevant).

The above comments are applicable to all sections in the paper – readers that are not fully updated in the nonlinear dynamic math jargon will not be able to follow parts of this paper.

---

## Author Comment (AC1) · 26 Feb 2017

The paper is dedicated to stability and bifurcation analysis of diffusion growth equation in its complete form, which includes curvature term and chemical composition term. Authors show that critical point on the Kohler curve is the point of saddle-node bifurcation. The time scale associated with this point which can be interpreted as time scale of aerosol activation is estimated analytically. Also the analysis of the more complicated case when the aerosol growth is accompanied by decrease of surrounding humidity has been done. The second saddle-node point was found in this analysis. At last the numerical calculation of hysteresis profiles of supersaturation and aerosol wet radius in adiabatic vertically-oscillating air parcel has been done. The article is interesting and I recommend it for publication

in NPG after two additions.

We thank the reviewer for the positive evaluation of the paper and the helpful comments addressed in detail below. We enclose a revised version of the paper.

> Firstly I recommend adding some elementary mathematical explanation what saddle-node bifurcation and cusp catastrophe are. May be authors will write a separate Appendix on this topic. This will help the reader to better understand the article.

Following the reviewer's advice, we have added relatively verbose explanations of the key nonlinear dynamics nomenclature used in the text. Instead of creating a separate appendix, the introductory material is included in a rewritten and substantially extended section on the saddle-node bifurcation. We have also added a new figure (Figure 1 in the revised manuscript) which is intended to be a graphical aid in interpretation the mathematical considerations implied in the fixed point analysis.

> Secondly I recommend to compare the results obtained in Sections 6, 7 and Fig.4 with results of study by Pinsky et al., (2013: J. Atmos. Sci., 70, 2778-2793). In this study some analytical investigation of monodisperse droplet spectra evolution as well as parcel model investigation are carried out. Possibly the matching of equations from this study and the ones in the reviewed one can bring some new results.

We have added a reference to the suggested work when mentioning studies that addressed the hysteretic behaviour of the system due to the kinetic limitations. The focus of our work is on the hysteresis occurring close to equilibrium – at negligible vertical velocities (small fractions of centimetres per second, if an air-parcel framework is used).

We have not found a direct link between the phase-relaxation timescales derived in the study of Pinsky et al. with the activation timescale derived herein.

---

## Author Comment (AC2) · 26 Feb 2017

The authors discuss cloud droplet activation and deactivation in terms of saddle-node bifurcation. By doing so they can employ a nonlinear dynamic approach to study the properties of such processes. Specifically, they approximate the drop growth by diffusion equation to a normal form of a saddle-node bifurcation (df/dt = a + fȨ̈2) and therefore they could use the properties of such form to study hysteresis and catastrophe behavior. The approach and the mathematical insights are very interesting but in order to make this paper accessible to readers from cloud physics, the authors should be much more generous in the details they provide in the mathematical derivations, and invest efforts in translating the mathematical insights to physical ones.

We thank the reviewer for the evaluation of the paper and the provided comments which we address in detail below. We enclose a revised version of the paper.

> A reader that is not fully updated with the jargon of nonlinear dynamics will find this paper hard to follow. The explanations in some sections are fully based on such jargon (see part 4 - "... the coalescence of the fixed points is associated with a passage throu

We have rewritten and significantly extended the section on saddle-node bifurcation in which the jargon is first used. The rewritten section contains explanations of the basic nomenclature.

We have not tried to refrain from using the jargon, though. The journal choice was motivated by the aim of addressing the nonlinear dynamics community as well.

> On the same note, throughout the paper the explanations are very slim. It starts from the overview in the introduction in which the Kohler theory is hardly mentioned (although it is central in the paper).

A paragraph introducing and pointing out the role of the Köhler theory in the presented mathematical model was added. While we fully agree that it is central in context of atmospheric CCN, the presented analysis – in principle – applies to activation phenomena in a wider context. This is now also highlighted in the introducing as well as in the paragraph where the Köhler theory is first mentioned.

> Moreover, many of the numbers provided there are not so accurate. One can have coarse mode aerosols larger than a micron. Concentration can vary between 10's to 10000's etc.

We have extended the given ranges following the suggestion of the reviewer, at the same time changing the numbers into more approximate textual representation ("from tens to thousands", "from fractions of to multiple micrometres") – in an attempt to underline the roughness of the estimation.

> In the next chapters where they develop the mathematical framework, they should add guidelines and physical insights in each of the main steps. What does it mean Saddle-node bifurcation at Köhler curve maximum? They should explain in not from a mathematical point of view (" ... when the fixed points coalesce into a half-stable fixed point ... ") but from a physical point of view. What is the meaning of this point. What can we learn about it from the Köhler theory? This is true to all mathematical steps in the paper.

We have added several sentences addressing this point. In particular, it is now clearly stated how stability of the fixed points relates to CCN growth and activation.

We have added a new figure (Fig. 1 in the revised manuscript) aimed at depicting the consecutive steps taken in the fixed point analysis as well as depicting how the phase portrait of the system can be recognised in the flipped Köhler curve.

> While the mathematical derivations look right (as much as I checked) the math derivations details are slim and there is hardly no physical interpretation to the shown insights (which could make this paper much more relevant).

While we hope that the introduced changes improved the text, let us also point out that throughout the paper we have in fact openly acknowledged that the discussed mathematical nuances of the studied system likely have limited relevance to cloud phenomena, most notably due to the monodisperse assumption ("limitations certainly
restrain the relevance of the presented calculations to real-world problems", "[close-to-equilibrium hysteresis] of no foreseeable relevance to the macroscopic behaviour of the large-scale cloud systems"). Yet, as underlined in the abstract and conclusions, the employed approximations and the depicted hysteretic behaviour are of relevance in construction of numerical schemes for solving drop growth equations.

Nevertheless, we consider the derived analytical estimate of the activation timescale readily applicable in studies dealing with cloud microphysics. This has been clearly pointed out in the abstract and conclusions.

> The above comments are applicable to all sections in the paper – readers that are not fully updated in the nonlinear dynamic math jargon will not be able to follow parts of this paper.

In a new last paragraph of the introduction, we have acknowledged appearance of the jargon throughout the text, and referred the reader to selected introductory chapters in the book of Strogatz for reference. Nevertheless, we do hope the rewritten section on fixed point analysis, supplemented with the new figure, makes the paper much more approachable.

---

## Author Response (AR1)

[revised manuscript text omitted]

Results of numerical simulations discussed in section 7.

---

## Referee Report (RR1)

Review of "On the CCN [de]activation nonlinearities" by Sylwester Arabas and Shin-ichiro Shima

The study investigates the activation and deactivation of aerosols, a fundamental process in cloud physics, using the mathematical framework of nonlinear dynamics. In doing so, the authors also introduce this methodology to the cloud physics community. Considering myself a member of the latter group, I found the references to classical textbooks an essential prerequisite for approaching this study. Of course, this journal is dedicated to nonlinear processes in geophysics but this study's subject is of interest for a broader audience. Therefore, some more physical explanations accompanying the mathematical framework of nonlinear dynamics might be necessary to make this study more approachable, especially in Section 7. Moreover, there are some more minor points and technical flaws which should be fixed easily.

All in all, the study is interesting, offers a new point of view, and is, therefore, worth publication after some very minor revisions.

Very minor comments:
- Physical interpretations: Reducing the discussion of Fig. 5 to terms like hysteresis, cusp bifurcation, and catastrophe makes anyone who is less familiar with nonlinear dynamics feel uncomfortable. It should be easily possible to associate the observed behavior with physical timescales as the activation timescale (see Fig. 3 of the manuscript), the phase relaxation timescale (e.g., Eq. (17) of Korolev and Mazin, 2003, JAS) or the evaporation timescale (e.g., Eq. (2) of Lehmann et al., 2009, JAS).
- Description of numerical model: Is system (17) complete? There should be a prognostic equation for the ambient vapor density $\rho_v$. If not, how is the supersaturation calculated? Are all equations of (17) solved with the same time step?
- Implications for modeling: Although I think the advices regarding the numerical solution of the activation/deactivation process are of major importance, I feel that there should be some more text on it in the introduction of the manuscript. Otherwise, the switch to the discussion of the model timestep in Section 7, line 348 – 359, feels too abrupt. Similarly, the term "stiffness" is mentioned first in line 365, but could be mentioned earlier (e.g., Section 2) to introduce the reader earlier to the numerical problems in the modeling of activation/deactivation.

Technical comments:
- Mathematical equations are a part of a sentence. Therefore, punctuation should also be considered in equations.
- There is a wide variety of notations used for derivations. Newtonian (Eq. 1, 2, 5, 7, 8, 9, 11), Leibnizian (Eq. 13), and Lagrangian (Eq. 12). Please stick to one.

---

## Author Response (AR2)

**Reply to the comments of Anonymous Referee #3**

We thank the Referee for the helpful remarks. We are glad the general impression was very positive. In what follows we address the specific points in the review, commenting on the related changes in the manuscript. A version of the manuscript with all changes highlighted is enclosed.

> Physical interpretations: Reducing the discussion of Fig. 5 to terms like hysteresis, cusp bifurcation, and catastrophe makes anyone who is less familiar with nonlinear dynamics feel uncomfortable. It should be easily possible to associate the observed behavior with physical timescales as the activation timescale (see Fig. 3 of the manuscript), the phase relaxation timescale (e.g., Eq. (17) of Korolev and Mazin, 2003, JAS) or the evaporation timescale (e.g., Eq. (2) of Lehmann et al., 2009, JAS).

While we agree that the embraced approach for analysis of the drop growth system can be used to discuss other timescales in the system, we argue that introducing it would not fit into a minor revision of the paper requested by the editor. Section 7 describing the numerical simulations is included in the paper with the aim of confirming that the hysteretic phenomena predicted through analyses of a simplified system are traceable also when the simplifying assumptions are lifted. The discussion of Fig. 5 is thus intentionally limited to the aspects previously discussed in the text.

> Description of numerical model: Is system (17) complete? There should be a prognostic equation for the ambient vapor density $\rho_v$. If not, how is the supersaturation calculated? Are all equations of (17) solved with the same time step?

The system is complete. Ambient vapour density (and hence supersaturation) can be diagnosed from the initial vapour content and the wet radius $r_w$ from the mass conservation: $\rho_v + N\rho_w \frac{4}{3}\pi r_w^3 = \text{const}$. All equations are solved with the same timestep. Two sentences clarifying it were introduced into the text (one in section 6, one in section 7).

> Implications for modeling: Although I think the advices regarding the numerical solution of the activation/deactivation process are of major importance, I feel that there should be some more text on it in the introduction of the manuscript. Otherwise, the switch to the discussion of the model timestep in Section 7, line 348 – 359, feels too abrupt. Similarly, the term "stiffness" is mentioned first in line 365, but could be mentioned earlier (e.g., Section 2) to introduce the reader earlier to the numerical problems in the modeling of activation/deactivation.

The issue of stiffness is expected to appear when largely different temporal/spatial scales are present in the system. It is thus not relevant to the monodisperse case which is discussed in the preceding sections, and mentioning it earlier in the text could be misleading, in our opinion. For clarity, the sentence mentioning stiffness in section 7 begins now with an explanation of the causes of the stiffness. Following the suggestion of the reviewer, we have extended the introduction with a description of the structure of the note and its rationale.

> Mathematical equations are a part of a sentence. Therefore, punctuation should also be considered in equations

Punctuation around numbered equations has been corrected.

> There is a wide variety of notations used for derivations. Newtonian (Eq. 1,2,5,7,8,9,11), Leibnizian (Eq. 13), and Lagrangian (Eq. 12). Please stick to one.

The Leibnizian notation was replaced with Newtonian in Eq. (13). Introducing the Lagrangian notation in Eq. (12) seems justified, in our opinion, as it is the only case where the derivative is taken with respect to radius rather than time.

[revised manuscript text omitted]
 super-saturation in convective clouds, J. Meteorol. Soc. Jpn., 28, 122–130, 1950.